# LAM3D: Large Image-Point-Cloud Alignment Model for 3D Reconstruction from Single Image

**Ruikai Cui**[1] [*]    **Xibin Song**[2] [†]    **Weixuan Sun**[2]    **Senbo Wang**[2]    **Weizhe Liu**[2]
**Shenzhou Chen**[2]    **Taizhang Shang**[2]    **Yang Li**[2]    **Nick Barnes**[1]    **Hongdong Li**[1]    **Pan Ji**[2]

[1]Australian National University    [2]Tencent XR Vision Labs

## Abstract

Large Reconstruction Models have made significant strides in the realm of automated 3D content generation from single or multiple input images. Despite their success, these models often produce 3D meshes with geometric inaccuracies, stemming from the inherent challenges of deducing 3D shapes solely from image data. In this work, we introduce a novel framework, the Large Image and Point Cloud Alignment Model (LAM3D), which utilizes 3D point cloud data to enhance the fidelity of generated 3D meshes. Our methodology begins with the development of a point-cloud-based network that effectively generates precise and meaningful latent tri-planes, laying the groundwork for accurate 3D mesh reconstruction. Building upon this, our Image-Point-Cloud Feature Alignment technique processes a single input image, aligning to the latent tri-planes to imbue image features with robust 3D information. This process not only enriches the image features but also facilitates the production of high-fidelity 3D meshes without the need for multi-view input, significantly reducing geometric distortions. Our approach achieves state-of-the-art high-fidelity 3D mesh reconstruction from a single image in just 6 seconds, and experiments on various datasets demonstrate its effectiveness.

## 1   Introduction

High-quality 3D mesh creation has drawn increasing attentions for its great potential in numerous fields, such as video games, virtual reality, and films. Traditionally, 3D assets are usually created manually by expert artists or alternatively reconstructed from multi-view 2D images, which are time-consuming. Recent proposed Large Reconstruction Models (LRMs) [15, 62, 48, 52] from a single view or few-shot images bring significant advancements for high-fidelity 3D content creation.

Single view based LRMs [15, 13] take a single-view image as input and use transformers to directly regress tri-plane-based [3] neural radiance fields (NeRF) [31] for 3D content creation. However, as shown in Fig. 1 (b), due to ambiguity of 3D geometry from a single image, single-view-based LRMs commonly suffer from geometric distortions, especially for the unseen areas of the input view. Inspired by Hong *et al*. [15], LRMs with multi-view inputs [62, 48, 52, 20] are proposed, where they typically first feed the single-view image into multi-view image diffusion models [51, 43, 28] for few-shot multi-view images generation. Then multi-view tri-plane features are generated and converted into 3D representations (NeRF [31], SDFs [34], *etc*.) for 3D mesh creation. However, as shown in Fig.1 (c), the generated multi-view images lack guaranteed multi-view consistency, which can lead to geometric distortions.

To relieve these problems, we propose to introduce a 3D point cloud prior for single image based 3D mesh reconstruction with Large Image-Point-Cloud Alignment Model (LAM3D). LAM3D aims to

---

[*]The contribution of Ruikai Cui was made during an internship at Tencent XR Vision Labs.
[†]Corresponding author.

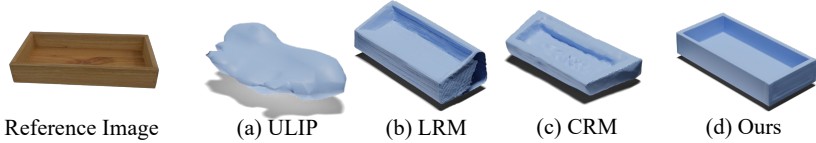

| Reference Image | (a) ULIP | (b) LRM | (c) CRM | (d) Ours |

Figure 1: An example of single-image reconstruction from state-of-the-art methods: (a) ULIP [63], (b) LRM [13], (c) CRM [54], and (d) Ours (LAM3D).

transfer single image features into point cloud features, enriching the image features with accurate and effective 3D information for high-fidelity 3D mesh reconstruction. The motivation behind this choice is obvious: 3D point cloud contains effective structure prior of 3D geometries. Meanwhile, approaches [63, 64] of modality feature alignment have been proposed, aiming to learn unified representations of language, images, and point clouds for 3D understanding. Nevertheless, as shown in Fig. 1 (a), these techniques fall short when they use the aligned image features to replace point clouds features for 3D mesh creation. This is largely due to the exclusive usage of contrastive loss [63, 64] and the aligned feature size is commonly with $(512 \times 1)$[63] and $(1024 \times 1)$[64], which is inadequate to fully capture the rich diversity of 3D shapes.

In this work, we extend the image and point cloud alignment for 3D mesh creation, and present the Large Image-Point-Cloud Alignment Model (LAM3D) which utilize the prior of 3D point cloud for high-fidelity feature alignment. We start by designing an effective point cloud-based network with hierarchical tri-planes [3] for 3D mesh reconstruction. Following this, taking single image as input, image features are extracted using DINO [2], whilst LAM3D is employed to transpose these image features into tri-planes obtained via the point-cloud-based network. The tri-plane representation [3] allows for the tuning of image features with precise and effective 3D information, which is essential for high-fidelity 3D mesh reconstruction. Meanwhile, to better retain 3D structure information and reduce the interference of each plane (XY, XZ, YZ) in tri-plane features, we enlist the use of independent diffusion operations to transfer image features to each respective plane. Note that the transferred features are with size of $3 \times C \times M \times N$ ($(3 \times 2 \times 32 \times 32)$ in our experiments), which is larger than previous representation size of $(512 \times 1)$[63] and $(1024 \times 1)$[64], thus contains more information for better 3D mesh creation. As shown in Fig. 1 (d), more reasonable geometric mesh can be reconstructed by our approach.

The main contributions of our work are as follows:

- We propose an effective Large Image-Point-Clouds Alignment Model (LAM3D) for 3D mesh reconstruction from a single image. By utilizing point cloud priors, LAM3D delivers precise image tri-plane feature transformation, thereby enhancing the quality and accuracy of 3D mesh creation.

- To preserve 3D structural information more effectively, we employ independent diffusion processes that transmit image features to each respective tri-plane (XY, XZ, and YZ) for accurate 3D mesh reconstruction.

- Our method significantly relieves geometric distortions caused by ambiguities of 3D shape inferred from the image modality, and experiments on various of dataset demonstrate the effectiveness of our approach.

## 2 Related Works

**Single Image to 3D Reconstruction:** Recently, LRM [15] was the first to demonstrate that a regression model can be trained to robustly predict tri-plane-based NeRF from a single-view image. Inspired by LRM, numerous large models[25, 26, 36, 54, 69, 53, 50] for 3D reconstruction have since been proposed. Among these, Instant3D [20] trains a text-to-few-shot multi-view image diffusion model, as well as a multi-view LRM, to enable fast and diverse text-to-3D generation. Subsequent works have extended LRM by incorporating pose prediction from multi-view images [52], combining it with diffusion [62], and specializing on human data [56]. Tang et al.[48] introduced the Large Multi-View Gaussian Model to generate high-resolution 3D models from text prompts or single-view images. Wang et al.[54] presented a single-image-to-3D generative model that generates six orthographic images, utilizing Flexicubes [42] as a 3D representation. Concurrently, Wu et al. [58]

proposed Direct3D, a method that differs from ours primarily in its approach to feature extraction from point clouds. Direct3D utilizes learnable tokens for feature extraction, whereas our method introduces a projection module to explicitly convert point cloud features into a tri-plane structure.

**Generative Modeling of 3D shapes:** In recent years, a multitude of studies have been dedicated to the generation of 3D shapes. Existing 3D generative models have been developed using a variety of frameworks, including generative adversarial networks (GANs)[57, 1, 67], variational autoencoders (VAEs)[47, 11, 17, 32, 12], normalizing flows [65, 32], autoregressive models [45, 32], energy-based models [61, 8], and more recently, denoising diffusion probabilistic models (DDPMs) [30, 68, 27, 49, 24, 4, 44, 60]. Luo *et al*. [30] pioneers the application of DDPMs for modeling the distribution of raw point clouds. Fellowing Luo *et al*, several works [49, 24, 4] explore generative modeling of 3D shapes in latent space to reduce computational complexity and enhance generation quality. For instance, LION [49] proposed an effective latent-space diffusion model to generate novel point clouds, while 3DQD [24], SDFusion [4] utilize DDPM to model Signed Distance Function (SDF) in the latent space of an autoencoder for 3D shape generation. Concurrently, other research efforts have investigated DDPMs for 3D shape generation with other representations, such as mesh [27], occupancy grid [44], and neural radiance fields [33].

**Feature Alignment / Multimodal 3D:** Multimodal approaches are mainly about image-text and image-text-point-clouds modalities. Several works [21, 23, 46] propose to learn interactions between image regions and caption words using transformer-based architectures, which show great predictive capability despite being costly to train. Meanwhile, CLIP [38] uses image and text encoders to output a single image/text representation for each image-text pair, and then aligns the representations to a unified latent space. Built upon CLIP [38], ULIP [63] and its successor, ULIP2 [64], aim to learn a unified representation that integrates images, text, and 3D point clouds. These methods establish a 3D representation space aligned with the CLIP latent space by leveraging synthesized 3D-text-image triplets. The most closely related work to ours is Michelangelo [66], which aligns point cloud features with frozen CLIP image/text embeddings through an autoencoder. Unlike Michelangelo, which employs contrastive loss to learn an aligned latent space, our approach uses diffusion to align 3D and 2D modalities. This results in a more continuous latent space, beneficial for 3D reconstruction.

## 3 Method

In this section, we present the detailed design of our approach. The full training process consists of two stages: *Point Cloud Compression* (Sec. 3.1) and *Image-Point-Cloud Alignment* (Sec. 3.2).

### 3.1 Stage 1: Point Cloud Compression

As shown in Fig. 2, we encode point clouds into a tri-plane representation and then decode SDF values from the reconstructed tri-plane to extract 3D meshes via marching cubes [29]. The point cloud compression consists of *Initial Point Feature Extraction* and *Tri-plane Compression*. Specifically, we propose a point-cloud-based 3D reconstruction network to convert point cloud into a latent tri-plane, which can retain rich 3D information. In the tri-plane compression module, the latent tri-plane is up-sampled to reconstruct tri-planes followed by an MLP to compute SDF values. As shown in Fig. 2, hierarchical tri-plane are used, including latent tri-planes and reconstructed tri-planes, where the latent tri-plane is designed for better image-point-cloud feature alignment and the reconstructed tri-plane is for better mesh reconstruction.

### 3.1.1 Initial Point Feature Extraction

To compress a point cloud into latent tri-planes, we employ a transformer-based approach to learn long-range interactions between points. We first use Furthest Point Sampling (FPS) to downsample the input point cloud to $n$ center points. Then, we gather the K-nearest neighbors of the $n$ center points together as a small point cloud and use a shallow PointNet [35] to embed each small point cloud as an embedding. By doing so, we reduce the sequence length of the point cloud, and then we can use a transformer to process these embeddings.

After the transformer step, we need to collect the embeddings to form the three planes of the tri-plane structure. To this end, we use a projection operation. In specific, each embedding feature is associated with a knn center. For each plane $(XY, XZ, YZ)$, we diminish one axis of the center points and

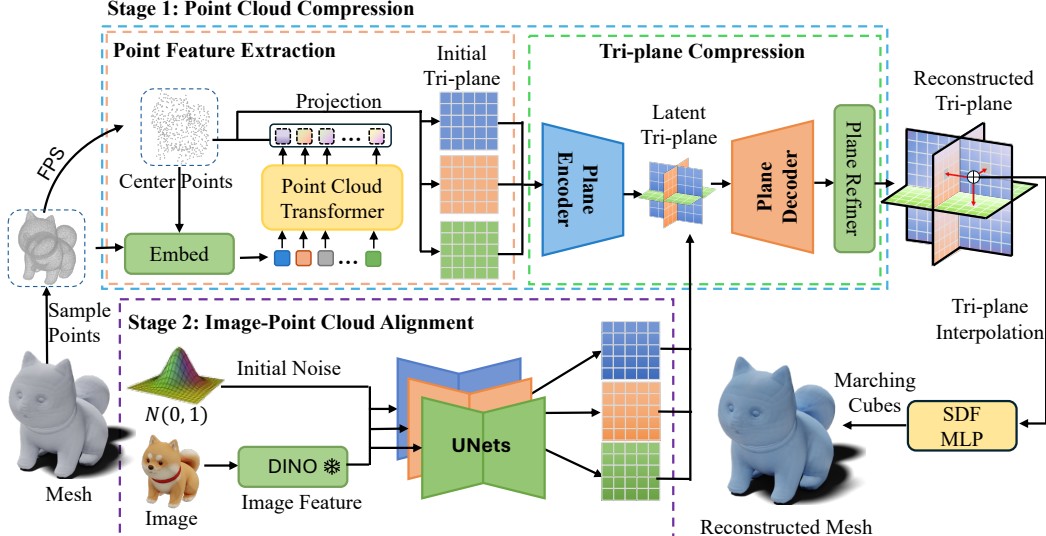

Figure 2: Overview of our method. Our method contains two training stage. **Stage 1**: we train an encoder-decoder structure to take point clouds as input and compress it to a latent tri-plane representation; **Stage 2**: we employ diffusion to align image modality to latent tri-planes obtained in stage 1. The diffusion step takes an initial noise and an image feature from a freezed DINO feature encoder and progressively align the image feature to the latent tri-plane. **Inference**: To reconstruct a 3D mesh from a single-view image, we use the alignment step, following the decoder (Plane Decoder, Plane Refiner) from the compression step, to predict a tri-plane. Then, we can use algorithms like marching cubes to extract 3D meshes from the reconstructed tri-plane.

voxelize the center points into a plane of size $\mathbb{R}^{C \times R \times R}$ where $C$ is the channel dimension that is the same with the embedding dimension, and $R$ is the plane resolution. We use average operation to aggregate embeddings associated with center points that land into the same voxel. As shown in Fig. 2, the point cloud is converted into a initial tri-planes that faithfully preserves 3D information.

### 3.1.2 Tri-plane Compression

The initial tri-planes is a sparse tensor as many of its grids are empty if there is no center points land there. To obtain a more compact and expressive latent tri-plane representation, we adopt an plane autoencoder to further compress initial tri-planes to latent tri-planes.

**Plane Encoder** We find that directly aligning image features with initial tri-planes leads to two drawbacks: (1) Aligning in high-resolution requires high computational complexity; (2) Empty grids and overwhelmed details in high-resolution tri-planes substantially increase training difficulty, leading to noisy results. To further compress the initial tri-planes into a more compact latent, we opt for a transformer-based plane encoder. It consists of a convolution layer to down-sample the initial tri-planes followed by successive transformer blocks to achieve inter-plane correlations. In each transformer block, we follow Cui *et al*. [6] to concatenate tri-planes into one sequence and adopt linear attention [37] to process the sequence to model inter-plane correlations. In our settings, the extracted initial tri-plane is down-sampled by 4 times to obtain a point cloud latent tri-plane $t \in \mathbb{R}^{(3 \times 2 \times 32 \times 32)}$.

**Plane Decoder** We introduce a hierarchical scheme to decode the latent tri-plane into a high-resolution reconstructed tri-plane, as depicted in Fig. 2, and for predicting the Signed Distance Function (SDF). Initially, the latent tri-plane is processed through a standard decoder that mirrors the encoder. This decoder comprises successive transformer blocks coupled with a convolutional layer, which augments the latent tri-plane by a factor of four.

**Plane Refiner** After Plane Decoder, we adopt an intra-plane refinement module to further enhance the reconstruction quality. In latent tri-plane generation, the original point clouds are condensed to form a compact tri-plane latent, the compression inaccuracies are unavoidable. Therefore, we propose

an asymmetric decoding of the tri-plane, which is accomplished by individually refining each plane with a plane refiner module subsequent to the conventional decoder. Specifically, the plane refiners are three small-scale UNet [41] models to refine each plane separately. We find that refining each plane individually extensively improves mesh quality while retains inter-plane correlations as we demonstrated in Appendices.Tab. 5. Finally, we can query SDF values of any spatial position using the SDF-MLP by interpolating features on the reconstructed tri-plane and extract 3D meshes using Marching Cubes [29].

### 3.1.3 Compression Loss Function

The training objective for point cloud compression comprises a KL penalty and a geometry loss. To align image features onto a 3D-aware latent space, we attempt to preserve 3D prior to the greatest extent, even in the most compact compressed tri-plane latent. However, we discovered that the compressed tri-plane latent fails to maintain the 3D prior, leading to artifacts in the mesh outcomes if we merely employ an arbitrary encoder to compress the initial tri-plane without explicit constraints. Hence, we propose to concurrently supervise the latent tri-plane $t$ and reconstructed tri-plane $P$ using the ground truth SDF and two distinct SDF MLPs ($\Phi_t$ and $\Phi_P$). To this end, we combine the geometric SDF loss and latent regularization to train the compression module in an end-to-end manner. Formally, to allow the predicted tri-plane recover SDF values, we use the geometry loss:

$$\mathcal{L}_{geo} = \mathcal{L}_{sdf} + \mathcal{L}_{normal}. \tag{1}$$

These two terms are defined as:

$$\mathcal{L}_{sdf} = \lambda_1 \sum_{p \in \Omega_0} ||\Phi_P(p)|| + \lambda_2 \sum_{p \in \Omega} ||\Phi_P(p) - d_p||, \tag{2}$$

$$\mathcal{L}_{normal} = \lambda_3 \sum_{p \in \Omega_0} ||\nabla_p \Phi_P(p) - n_p||, \tag{3}$$

where $\Omega_0$ and $\Omega$ are sets of query points sampled from object surface and off-surface points sampled from $[-1, 1]^3$. The grouth truth SDF values and surface normal are denoted as $d_p$ and $n_p$, respectively. The gradient $\nabla_p \Phi_P(p) = \left( \frac{\partial \Phi_P(p)}{\partial X}, \frac{\partial \Phi_P(p)}{\partial Y}, \frac{\partial \Phi_P(p)}{\partial Z} \right)$ represents the direction of the steepest change in SDF. It can be computed using finite difference, *e.g.*, the partial derivative for the X-axis component reads $\frac{\partial \Phi(p)}{\partial X} = \frac{\Phi_P(p+(\delta,0,0)) - \Phi_P(p-(\delta,0,0))}{2\delta}$ where $\delta$ is the step size. Similarly, the latent sdf loss $\mathcal{L}_{lsdf}$ is defined as:

$$\mathcal{L}_{lsdf} = \lambda_4 \sum_{p \in \Omega_0} ||\Phi_t(p)|| + \lambda_5 \sum_{p \in \Omega} ||\Phi_t(p) - d_p||. \tag{4}$$

We use the corresponding MLP to predict a SDF value from either latent tri-plane $t$ or the reconstructed tri-plane $P$. Specifically, given a query position $p$, we project it to each of the planes and retrieve the feature vectors $F_{xy}, F_{xz}, F_{yz}$ via bilinear interpolation. Then, the signed distance can be predicted as: $\Phi(p) = \text{MLP}(F_{xy} \oplus F_{xz} \oplus F_{yz})$, where $\oplus$ denotes element-wise summation.

Overall, the training objective is defined as fellow:

$$\mathcal{L}_{comp} = \mathcal{L}_{geo} + \mathcal{L}_{lsdf} + \mathcal{L}_{KL}, \tag{5}$$

where $\mathcal{L}_{KL}$ is a KL penalty which leans towards a standard normal distribution on the learned latent tri-plane, akin to a Variational Autoencoder [19].

### 3.2 Stage 2: Image-Point-Cloud Alignment

We align image and point cloud features within the same feature space. Since the point clouds have been compressed into latent tri-planes with 3D-aware point cloud priors in stage 1, we can project the image features onto this 3D-aware latent. We experiment with two approaches to accomplish this goal. First, following reconstruction methods such as LRM [15], we can utilize a transformer-based structure to align the image features with the point cloud tri-planes, employing a distance metric like L1 or L2 loss. Alternatively, we can employ a probabilistic diffusion approach, wherein we initiate from random noise and denoise it, guided by the input image targeting the point cloud domain.

Our experiments reveal that the transformer-based model yields unstable results. This instability arises because 3D reconstruction from a single image is an ill-posed problem, as the image lacks information

about the occluded object parts. The 3D structural information in an image is not as rich as that in point clouds. Consequently, if we simply adopt a deterministic approach, the alignment result tends to be the average of all possible outcomes for regions that are not sufficiently observed [15]. Thus, we choose a diffusion model as a probabilistic alignment approach. Intuitively, the diffusion model can leverage its prior to envision unobserved regions and generate clear predictions for these areas, as demonstrated in previous 2D prior works [55, 22]. We provide more analysis in Appendices. D.2.

### 3.2.1 Alignment Model Structure

As shown in Fig. 2, first, we use a pre-trained DINO [2] image encoder to convert images to a $S \times D$ dim feature $z_{img}$ where $S$ is the squence length and $D$ is the feature dimension. Subsequently, the image features serve as conditioning information in the diffusion process, mapping the image feature from the image modality to the point cloud modality.

Unlike previous methods [5, 6, 12] that depend on a single diffusion model to map an arbitrary 1D ordering vector, we employ three parallel UNets to map the image feature to each of the three latent planes, separately. Since each tri-plane contains a 2D-image-like feature, as demonstrated in Appendices, Fig. 6. Prior methods [12, 6] use a single UNet to denoise concatenated planes. However, the UNet is designed without 3D-aware prior, meaning that there will be adjacent pixels without a direct relation due to the convolution operation being a local operation, leading to artifacts.

### 3.2.2 Alignment Loss Function

We utilize a diffusion model to align the image and the point cloud features. Diffusion Models [14, 40] are designed to learn a data distribution $q(z_0)$ by progressively denoising a normally distributed variable. The learning process is equivalent to performing the reverse operation of a fixed Markov Chain with a length of $T$, which transforms latent $z_0$ into purely Gaussian noise $z_T \sim \mathcal{N}(0, I)$ over $T$ time steps. The forward step in this process is defined as:

$$q(z_t|z_{t-1}) = \mathcal{N}(z_t; \sqrt{1 - \beta_t}z_{t-1}, \beta_t I), \tag{6}$$

where noisy variable $z_t$ is derived by scaling the previous noise sample $z_{t-1}$ with $\sqrt{1 - \beta_t}$ and adding Gaussian noise following a variance schedule $\beta_1, \beta_2, \ldots, \beta_T$.

We train three parallel diffusion models on each of the three planes by learning to reverse the above diffusion process. To achieve this, we adopt the approach proposed by Aditya *et al.* [39], wherein we use three neural networks that takes $z_t$ to directly predict $z_0$ corresponding to XY, XZ, and YZ planes. Specifically, given a uniformly sampled time step $t$ from the set $\{1, ..., T\}$, we generate $z_t$ by sampling noise from the input latent tri-plane $z_0$. A time-conditioned denoising autoencoder consisted of three parallel UNets [40], denoted by $\Psi$, is employed to reconstruct $z_0$ from $z_t$ given the alignment source, *i.e.*, the image latent feature $z_{img}$. The objective of the alignment is given by:

$$\mathcal{L}_{align} = ||\Psi(z_t, z_{img}, \gamma(t)) - z_0||^2, \tag{7}$$

where $\gamma(\cdot)$ represents a positional embedding and $|| \cdot ||^2$ denotes the mean squared error (MSE) loss.

## 4 Experiments

**Dataset:** Following previous works [54, 15], our model is trained on a subset of Objaverse [9] with 140k filtered 3D assets, and evaluated on unseen Objaverse objects and Google Scanned Objects [10]. The Objaverse dataset contains over 800k 3D assets. It was filtered to remove objects with thin faces and repeated/similar buildings, resulting in a final training set of 140k objects. More details can be found in Appendices. B.

**Baselines:** To assess the effectiveness of our approach, we compare our method with state-of-the-art approaches, including One-2-3-45 [25], LRM [15], SyncDreamer [26], Wonder3D [28], Magic123 [36], TGS [69], LRM [48] and CRM [54].

### 4.1 Implementation Details

During training, we employ a two-stage training pipeline. In stage 1, a point cloud compressor is trained, comprising a transformer-based point feature extractor and a tri-plane autoencoder. Given

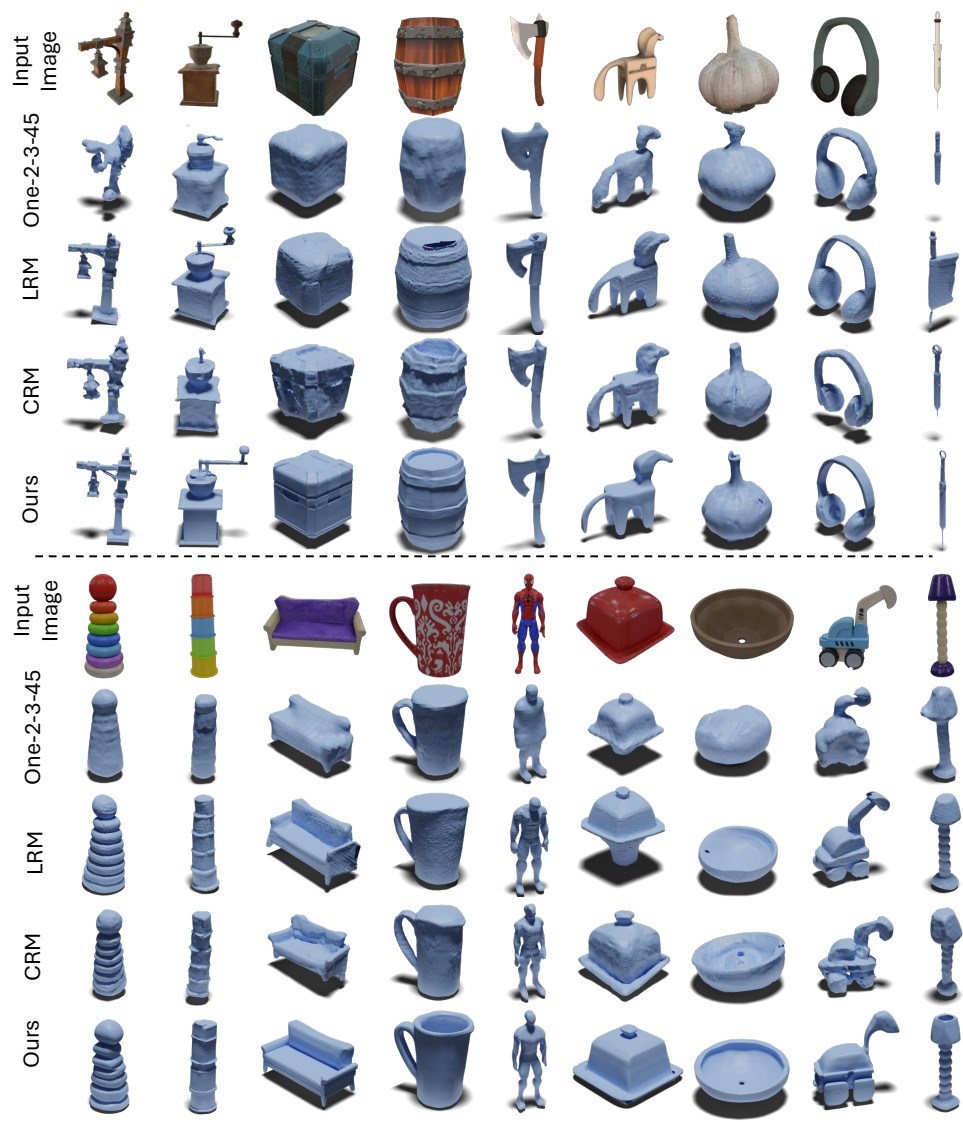

Figure 3: Rendered images of shapes reconstructed by various methods from single images. The upper samples are from Objaverse and the lowers are from Google Scanned Objects.

a point cloud, we sample 8192 points to feed them into the point feature extractor followed by a KNN-based sampler to obtain 512 point features. Subsequently, the point features are projected onto three orthogonal planes to generate a tri-plane with a $128 \times 128$ spatial resolution. Then we compress the tri-plane into a latent tri-plane of size $(3 \times 2 \times 32 \times 32)$ which has a feature dimension of 2. All components including the point feature extractor, tri-plane autoencoder, and two SDF MLPs, are trained in an end-to-end manner. We utilize 32 NVIDIA V100 GPUs to train for 20 epochs with a batch size of 25 and Adam [18] optimizer, which takes approximate 3 days.

In stage 2, the compressed latent tri-planes are utilized as supervision signals to train our alignment network. For an input image of size $(512 \times 512)$, we use a pre-trained DINO [2] to extract image feature of size $1025 \times 768$, where 768 is the feature dimension and 1025 is $32 \times 32$ image patches plus 1 global image feature. The image features serve as conditioning signal through cross-attention, guiding the diffusion process of the three UNets to generate three planes. During the alignment training, only the UNets are trained, while the DINO remains fixed. The training cost for the alignment is relatively low since we can pre-encode all the point clouds as well as all the source images. We employ 8 V100 GPUs to train the second stage model for 2 days with a batch size of 64.

Table 1: Quantitative comparisons for the geometry quality between our method and baselines.

| Method | Chamfer Dist.↓ | Vol. IoU↑ | F-Sco.(%)↑ |
|---|---|---|---|
| One-2-3-45 [25] | 0.0172 | 0.4463 | 72.19 |
| SyncDreamer [26] | 0.0140 | 0.3900 | 75.74 |
| Wonder3D [28] | 0.0186 | 0.4398 | 76.75 |
| Magic123 [36] | 0.0188 | 0.3714 | 60.66 |
| TGS [69] | 0.0172 | 0.2982 | 65.17 |
| OpenLRM [15, 13] | 0.0168 | 0.3774 | 63.22 |
| LGM [48] | 0.0117 | 0.4685 | 68.69 |
| CRM [54] | 0.0094 | 0.6131 | 79.38 |
| Ours | **0.0083** | **0.6235** | **85.40** |

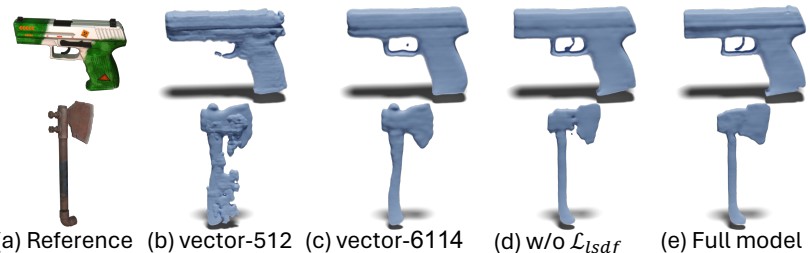

(a) Reference  (b) vector-512  (c) vector-6114  (d) w/o $\mathcal{L}_{lsdf}$  (e) Full model

Figure 4: Qualitative comparisons of different latent representations.

The inference stage requires only a single-view image of an object. The process starts by encoding this image into a feature representation using the DINO image encoder. This image feature is then aligned to a latent tri-plane representation through three independent diffusion UNets. The latent tri-plane is subsequently upsampled to a high-resolution tri-plane using the plane decoder and refiner from the first training stage. We then employ the SDF MLP to decode the signed distance for any query position on the tri-plane, which allows us to use the Marching Cubes algorithm to reconstruct a mesh from the tri-plane.

## 4.2 Comparisons with state-of-the-arts

**Qualitative Results:** Fig. 3 presents a qualitative comparisons between our approach and other state-of-the-art approaches including One-2-3-45 [25], LRM [15] and CRM [54]. Since LRM is not open-sourced, we use OpenLRM [13], an open-sourced implementation of LRM for comparisons. For the other baselines, we use their official codes and model weights. As shown in Fig. 3. While impressive results can be obtained with these methods, geometric distortions often occur due to ambiguities caused by single image or multi-view inconsistency. Additionally, these approach commonly fail for unseen areas of the input image, such as the inside of a cup. In contrast, our method leverages point cloud priors to reconstruct 3D meshes with better geometry and more details than all other baselines. Moreover, our model does not rely on multi-view consistency, unlike One-2-3-45, LRM, and CRM, thanks to the point cloud prior. This results in significantly reduced geometric distortions, as seen in the box, sofa and bowl examples in Fig. 3.

**Quantitative Results:** following previous studies [54, 48, 15], we use the Google Scanned Objects (GSO) dataset [10] to evaluate the effectiveness of our method. In line with CRM [54] and One-2-3-45 [25], we use Chamfer Distance (CD) [59, 7], Volume IoU and F-Score with a threshold of 0.05 to evaluate the mesh geometry. The results are shown in Tab. 1. It can be seen that our method outperforms all of the baselines on all the evaluation metrics. This demonstrates the effectiveness of our method for 3D reconstruction. Notably, our model generates mesh within only 6 seconds on an NVIDIA V100 GPU.

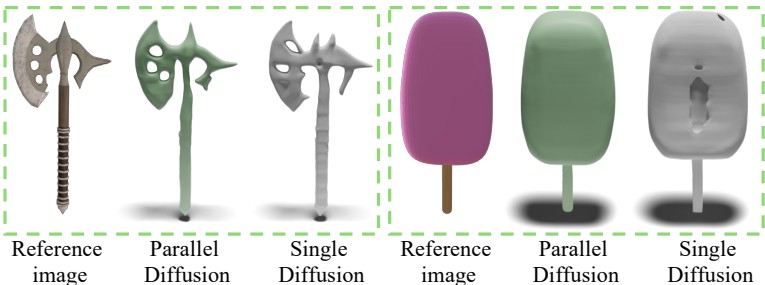

| Reference image | Parallel Diffusion | Single Diffusion | Reference image | Parallel Diffusion | Single Diffusion |

Figure 5: Green objects are generated from parallel UNets and gray samples are from single UNet.

## 4.3 Ablation Study

For our ablation study, we use a small subset of the Objaverse dataset containing 3000 game assets (weapons, gear, *etc.*) for training, and 216 objects for evaluation.

**Effectiveness of latent tri-plane representation:** In our method, we compress the point cloud as a latent tri-plane representation, which is then used for image-point-cloud feature alignment. In this study, we evaluate the effectiveness of this proposed latent tri-plane representation.

We compare three different variants in our ablation study: a) latent tri-plane w/ $\mathcal{L}_{lsdf}$: our introduced hierarchical decoding strategy, constraining latent tri-plane with a SDF loss; b) latent tri-plane w/o $\mathcal{L}_{lsdf}$, latent tri-plane without the SDF loss; c) *vector*, a vector with dim ($6114\times1$) (same size with ($3\times2\times32\times32$)) is used to represent the point cloud latent. We train the stage 1 compression models for point cloud-based reconstruction. As shown in Tab. 2, simply using a latent vector representation without spatial structure does not perform well with a performance of 4.81. However, converting the vector representation to tri-plane improves the performance to 1.95. Moreover, tri-plane with latent SDF loss (Ours in Tab. 2) can further refine the final results to 1.81. This emphasizes the importance of retaining 3D-aware structure in the point cloud latent representation and the necessary of introducing the latent SDF loss $\mathcal{L}_{lsdf}$.

In Fig. 4, we provide illustrations of different sizes and representations. We observe that a vector size of 512 performs poorly, which is inadequate to represent the 3D shapes. A vector with dimensions of $6114 \times 1$ can recover the structure of a 3D mesh, but details such as the trigger are missing. In Fig. 4 (d), we show that tri-plane structure can improve the visual performance with better trigger. In Fig. 4 (e), further refined results with better details can be obtained, validating the effectiveness of the latent SDF loss.

Table 2: Comparisons of different representations.

| Method | CD($\times10^4$)↓ |
|---|---|
| ULIP [63] | 35.23 |
| *vector* ($6114\times1$) | 4.81 |
| latent tri-plane w/o $\mathcal{L}_{lsdf}$ | 1.95 |
| latent tri-plane w/$\mathcal{L}_{lsdf}$ (Ours) | **1.81** |

Table 3: Quantitative comparison between parallel diffusion UNet and single UNet.

| Method | Single Diffusion | Parallel Diffusion |
|---|---|---|
| CD($\times10^3$)↓ | 3.92 | 2.59 |

**Image-point-cloud alignment model vs. ULIP:** In order to validate the effectiveness of our proposed image-point-cloud alignment model, we compare our method with ULIP [63], which is designed for point cloud feature alignment with text and images.

*ULIP-based image reconstruction:* ULIP uses a pre-trained CLIP and point cloud encoder to encode text, images, and point clouds to a feature of size ($512\times1$). Contrastive loss is utilized for aligning the features of point clouds, text, and images. To evaluate the performance of ULIP for 3D reconstruction, we design a decoder network that decodes the ($512\times1$)-dimension point cloud feature vector to a 3D mesh. We then use the decoder network to directly replace the point cloud features with the aligned image feature obtained by ULIP for 3D mesh reconstruction. As shown in Tab. 2, results obtained by ULIP [63] are unstable, leading to worse performance in charmer distance, which proves that the scheme of feature alignment used in ULIP [63] works poor for 3D reconstruction. On contrary, our method can obtain better results.

**Parallel Diffusion vs. Single Diffusion:** To better retain 3D structure information and reduce the interference of each plane (XY, XZ, YZ) in latent tri-planes, we enlist the use of independent parallel diffusion operations to transfer image features to each respective plane. We valid the effectiveness of using three parallel diffusion operations to do the feature alignment instead of using a single UNet in Tab. 3. Parallel UNet variant has smaller CD error, and we show qualitative comparisons in Fig. 5. We can observe that, thanks to the parallel UNet design, our model can focuses on aligning image feature to a specific plane and thus presents better visual quality. For example, the surface of the ice cream bar is broken in the reconstruction of the single UNet variant, while ours can maintains the continuity of the surface. Notably, to compare equally, we reduce the parameters of parallel diffusion to guarantee the same parameter size of parallel diffusion and single diffusion.

We provide more ablation studies as well as more visualization results in the appendix.

## 5 Conclusions

In this work, we present the Large Image and Point Clouds Alignment Model (LAM3D), which harnesses the prior of 3D point clouds for high-fidelity 3D mesh creation. We design an effective point-cloud-based network to compress point cloud into compact and meaningful latent tri-plane. Subsequently, taking a single image as input, Image-Point-Clouds Feature Alignment is designed to transfer image features to latent tri-planes, enriching the image features with accurate and effective 3D information for high-fidelity 3D mesh reconstruction. We believe that LAM3D has potential to improve 3D content creation and assist the workflow of digital artists.

*Limitations and Future Work:* The main goal of the proposed LAM3D is geometry reconstruction but texture reconstruction is not included, so that our method cannot achieve textured mesh reconstruction. We will further extend LAM3D to geometric and texture reconstruction in the future.

*Broader Impact:* We hold the view that LAM3D has the potential to enhance 3D content creation and aid digital artists in their workflow. LAM3D was conceived with these applications in mind, and we hope it can evolve into a useful tool that reduce the labour cost in 3D asset production. While we do not foresee any immediate harmful uses for LAM3D, we believe it is crucial for users to exercise caution to minimize impacts, considering that the alignment framework can also be employed for harmful intents.

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

## A  Diffusion Models Formulation

Diffusion Models are probabilistic models designed to learn a data distribution $z_0 \sim q(z_0)$ by gradually denoising a normally distributed variable. This process corresponds to learning the reverse operation of a fixed Markov Chain with a length of $T$. The inference process works by sampling a random noise $z_T$ and gradually denoising it until it reaches a meaningful latent $z_0$. Denoising diffusion probabilistic model (DDPM) [14] defines a diffusion process that transforms latent $z_0$ to white Gaussian noise $z_T \sim \mathcal{N}(0, I)$ in $T$ time steps. Each step in the forward direction is given by:

$$q(z_1, ..., z_T|z_0) = \prod_{t=1}^{T} q(z_t|z_{t-1}) \tag{8}$$

$$q(z_t|z_{t-1}) = \mathcal{N}(z_t; \sqrt{1 - \beta_t} z_{t-1}, \beta_t I) \tag{9}$$

The noisy latent $z_t$ is obtained by scaling the previous noise sample $z_{t-1}$ with $\sqrt{1 - \beta_t}$ and adding Gaussian noise with variance $\beta_t$ at timestep $t$. During training, DDPM reverses the diffusion process, which is modeled by a neural network $\Psi$ that predicts the parameters $\mu_\Psi(z_t, t)$ and $\Sigma_\Psi(z_t, t)$ of a Gaussian distribution.

$$p_\Psi(z_{t-1}|z_t) = \mathcal{N}(z_{t-1}; \mu_\Psi(z_t, t), \Sigma_\Psi(z_t, t)) \tag{10}$$

With $\alpha_t := 1 - \beta_t$ and $\bar{\alpha} := \prod_{s=0}^{t} \alpha_s$, we can write the marginal distribution:

$$q(z_t|z_0) = \mathcal{N}(z_t; \bar{\alpha}_t z_0, (1 - \bar{\alpha}_t)I) \tag{11}$$

$$z_t = \bar{\alpha}_t z_0 + \sqrt{1 - \bar{\alpha}_t} \epsilon \tag{12}$$

where $\epsilon \sim \mathcal{N}(0, I)$. Using Bayes' theorem, we can calculate $q(z_{t-1}|z_t, z_0)$ in terms of $\tilde{\beta}_t$ and $\tilde{\mu}_t(z_t, z_0)$ which are defined as follows:

$$\tilde{\beta}_t := \frac{1 - \bar{\alpha}_{t-1}}{1 - \bar{\alpha}_t} \beta_t \tag{13}$$

$$\tilde{\mu}_t(z_t, z_0) := \frac{\sqrt{\bar{\alpha}_{t-1}} \beta_t}{1 - \bar{\alpha}_t} z_0 + \frac{\sqrt{\alpha_t}(1 - \bar{\alpha}_{t-1})}{1 - \bar{\alpha}_t} z_t \tag{14}$$

$$q(z_{t-1}|z_t, z_0) = \mathcal{N}(z_{t-1}; \tilde{\mu}_t(z_t, z_0), \tilde{\beta}_t I) \tag{15}$$

Instead of predicting the added noise as in the original DDPM, in this paper, we predict $z_0$ directly with a neural network $\Psi$ following Aditya *et al.* [39]. The prediction could be used in Eq. 14 to produce $\mu_\Psi(z_t, t)$. Specifically, with a uniformly sampled time step $t$ from $\{1, ..., T\}$, we sample noise to obtain $z_t$ from input latent vector $z_0$. A time-conditioned denoising auto-encoder $\Psi$ learns to reconstruct $z_0$ from $z_t$, guided by the image feature $z_{img}$. The objective of latent tri-plane diffusion reads

$$\mathcal{L}_{ldm} = \|\Psi(z_t, z_{img}, \gamma(t)) - z_0\|^2 \tag{16}$$

where $\gamma(\cdot)$ is a positional encoding function and $\| \cdot \|^2$ denotes the Mean Squared Error loss.

## B  Dataset Preparation

The Objaverse contains over 800k 3D assets. However, there are many low-quality 3D assets that are not suitable for training a 3D reconstruction, such as object with only thin faces. We filter those object by removing object with only thin faces, repeated and similar buildings and senses. Finally, we obtain 140k object for training.

For the first stage compression training, we sample data from the object surface. Given a 3D object mesh, we first pre-process it to ensure it forms a watertight shape using the method proposed by Huang *et al.* [16]. Then, we normalize the shape within the $[-1, 1]^3$ box. Subsequently, we randomly sample on-surface points from the shape surface and uniformly sample off-surface points with ground truth SDF values in the $[-1, 1]^3$ space. In addition to the signed distance supervision, we leverage normal directions as an extra guidance to achieve detailed surface modeling. Consequently, we also sample the normal vector for each on-surface point.

For the alignment stage training, we render each object from 48 fixed camera position equally distributed on a sphere covering the object. However, we do not use all 48 rendered images for the

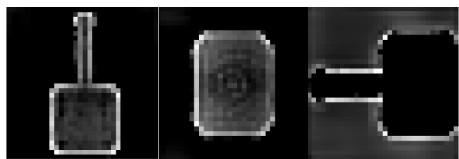

Figure 6: Visualization of latent tri-plane by converting the value to density.

second stage training. Instead, we count the area of observed region and only select the image with the largest observed area to align its feature to the point cloud modality at the second stage. This operation aims to reduce the ambiguity caused by symmetry of objects. By doing this operation, we also have the benefit that camera parameters are no longer needed to do the inference, the second stage model will determine this by itself in and end-to-end manner.

## C  Evaluation Metrics

We provide a detailed formulation of metrics that are not explicitly defined in the main manuscript to enhance understanding of our experimental outcomes

**Chamfer Distance (CD)**: This metric measures the average distance between two point sets, usually the ground truth and the predicted 3D reconstruction. Given two point sets $A$ and $B$, the Chamfer Distance is defined as:

$$CD(A, B) = \frac{1}{|A|} \sum_{a \in A} \min_{b \in B} ||a - b||^2 + \frac{1}{|B|} \sum_{b \in B} \min_{a \in A} ||a - b||^2, \tag{17}$$

where $|A|$ and $|B|$ denote the number of points in sets $A$ and $B$, respectively, and $||a - b||^2$ represents the squared Euclidean distance between points $a$ and $b$. Note that CD is a symmetric and non-negative distance measure.

To compute the CD for 3D reconstruction evaluation, we first sample 2048 points from the surfaces of the predicted mesh and the ground truth mesh. Then, we calculate the Chamfer distance between the point sets, as defined above.

**F-Score**: This metric is a measure of the trade-off between precision and recall, which are commonly used metrics for evaluating the quality of a 3D reconstruction. Given a threshold value $\tau$, the F-Score is defined as:

$$F(\tau) = 2 \frac{\text{precision}(\tau) \cdot \text{recall}(\tau)}{\text{precision}(\tau) + \text{recall}(\tau)}, \tag{18}$$

where $\text{precision}(\tau)$ and $\text{recall}(\tau)$ are the precision and recall values at the given threshold $\tau$, respectively. In our experiment, we set the threshold to 0.05 *w.r.t.* the Chamfer distance.

**Volume Intersection over Union (IoU)**: Volume Intersection over Union (IoU) is another metric used for evaluating the quality of 3D reconstructions. It measures the ratio of the volume of the intersection between the predicted mesh and the ground truth mesh to the volume of their union. The IoU is defined as:

$$IoU = \frac{\text{volume of intersection}}{\text{volume of union}} \tag{19}$$

To compute the IoU for 3D reconstruction evaluation, we first convert the predicted mesh and the ground truth mesh into Signed Distance Functions (SDFs) with a voxel resolution of $128 \times 128 \times 128$. All negative voxels in the SDFs represent the interior of the object. Then, we calculate the volume of the intersection and the volume of the union between the two SDFs. Finally, we compute the IoU using the formula above.

## D  More analysis

### D.1  Effect of latent tri-plane resolution and feature dimension

We conduct experiments to study the effect of latent tri-plane resolution and feature dimension. We design four compared variants of our full model. There are two aspects that needs to be considered, *i.e.*

Table 4: Quantitative comparison for the geometry quality between different spatial resolution and feature dimension of latent tri-plane on point cloud to 3D mesh.

| Spatial Reso. | Feature Dim. | #parameters | CD($\times 10^4$)$\downarrow$ |
|---|---|---|---|
| $8 \times 8$ | 2 | 384 | 4.15 |
| $16 \times 16$ | 2 | 1536 | 2.12 |
| $32 \times 32$ | 1 | 3072 | 1.87 |
| $32 \times 32$ | 2 | 6144 | **1.81** |
| $32 \times 32$ | 4 | 12288 | 2.63 |

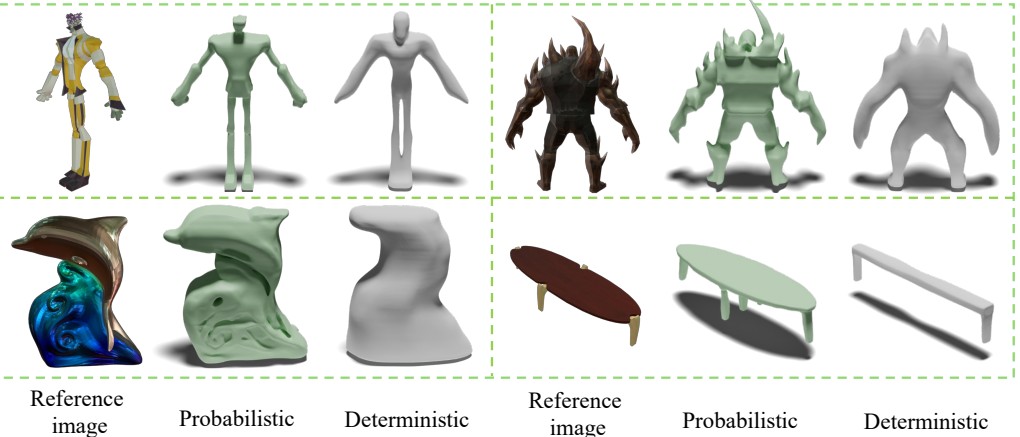

Reference image    Probabilistic    Deterministic      Reference image    Probabilistic    Deterministic

Figure 7: Alignment approach ablation study. We evaluate the single view reconstruction capability of deterministic vs. probabilistic approaches. Green objects are constructed from our probabilistic approach and gray samples are from a deterministic approach. We also present the reference image.

the spatial resolution and the feature dimension of each plane. We vary these parameters and evaluate the point cloud to mesh reconstruction capability in terms of CD. The result is presented in Tab. 4. From which we can observe that setting the spatial resolution and feature dimension as $(32 \times 32)$ and 2 are the best choice. We can also notice that, when we simply introduce a tri-plane structure, the reconstruction quality improved by 0.64 in terms of CD ($\times 10^4$) despite the number of parameters in the $2 \times 8 \times 8$ setting is significantly lower than it in the vector setting presented in Tab. 2. This improvement is brought by the tri-plane structure introduced 3D awareness to the latent structure, which benefits the decoding stage. Another founding is that improving the spatial resolution brings improvements to the reconstruction quality. However, the computation complexity will increase quadraticly as we increase the spatial resolution, therefore, increase the resolution beyond $32 \times 32$ becomes intractable in our experiment. Furthermore, we do ablation study on the feature dimension. We notice that increasing the feature dimension is not beneficial anymore, a potential reason is that increasing the feature dimension introduces too many parameters which leads to a latent space that is not compact enough to be continues. Therefore, it performs poorly when generalizing to novel objects. Also, decreasing the feature dimension to 1 leads to a worse performance as there is not adequate parameters to preserve the geometry information of the encoded object.

## D.2 Effect of using probabilistic alignment rather than deterministic

In terms of the modality alignment strategy, a navie way to address this is using a projection network that directly project an image feature to the point cloud feature modality. However, since this is an ill-pose problem and a single-view image can corresponding to many possible valid 3D shapes due to occlusion. A deterministic approach tends to produce blurry predictions [15]. We demonstrate such an effect by designing a deterministic Image-Point-Cloud alignment network. To this end, we follow LRM [15] and use a 12-layer transformer to directly align image feature to point cloud feature modality and train on the 140k Objavese dataset. Fig. 7 shows the difference between our probabilistic alignment approach compared with a deterministic method. From which we can observe that the reconstruction with probabilistic process involved produces more clear and sharp details than a deterministic approach.

Table 5: Quantitative comparison for the geometry quality between our full model w/ or w/o the plane refiner on point cloud to 3D mesh.

| Method | w/o Refiner | w/ Refiner |
|---|---|---|
| CD($\times 10^4$)↓ | 2.37 | **1.81** |

| Reference image | w/ refiner | w/o refiner |

Figure 8: We evaluate the single view reconstruction capability of our model w/ and w/o the plane refiner. Green objects are constructed from our model with the plane refiner and gray samples are from a the model without the plane refiner. We also present the reference image.

### D.3   Effect of plane refiner

To evaluate the effectiveness of our plane refiner module, we design a comparison about our full model with plane refiner and a variant without plane refiner. The result is shown in Tab. 5, and we also present a qualitative result in Fig. 8. We can observe that the plane refiner module improves the capability of our compression model, allowing it to recover more fine-grained details.

## E   More Visualization

We present more visualization for 3D reconstruction by aligning image and point cloud modality using our proposed method in Fig. 9 and Fig 10.

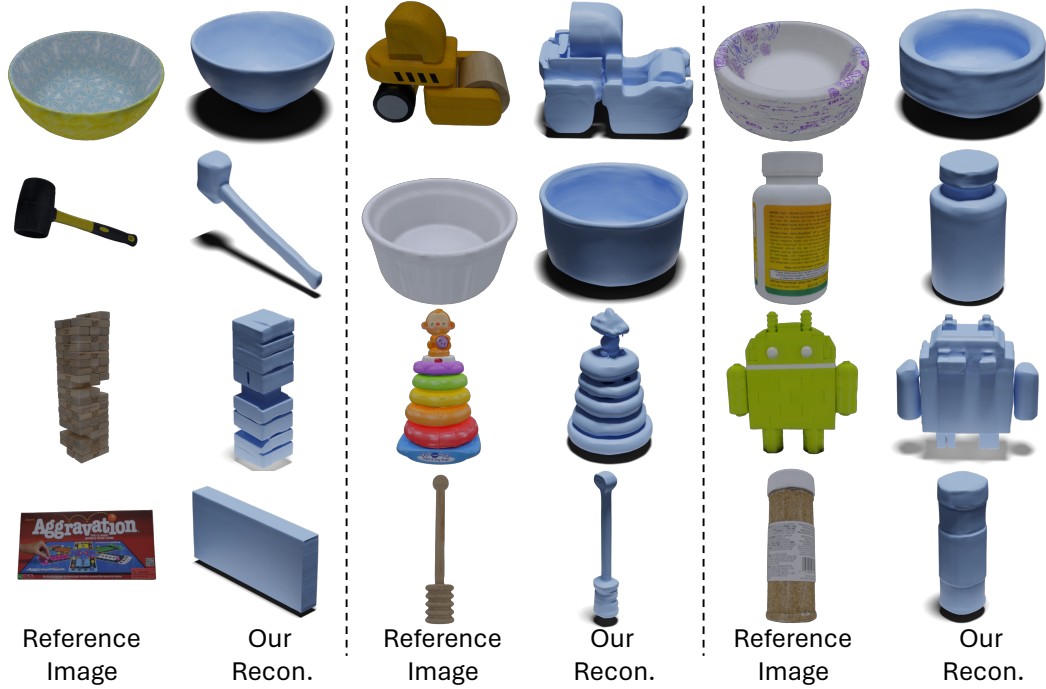

| Reference Image | Our Recon. | Reference Image | Our Recon. | Reference Image | Our Recon. |

Figure 9: Rendered images of shapes reconstructed by our LAM3D from single images on the GSO dataset. For each tuple of samples, the left image is the reference image and the right image is the reconstructed geometry.

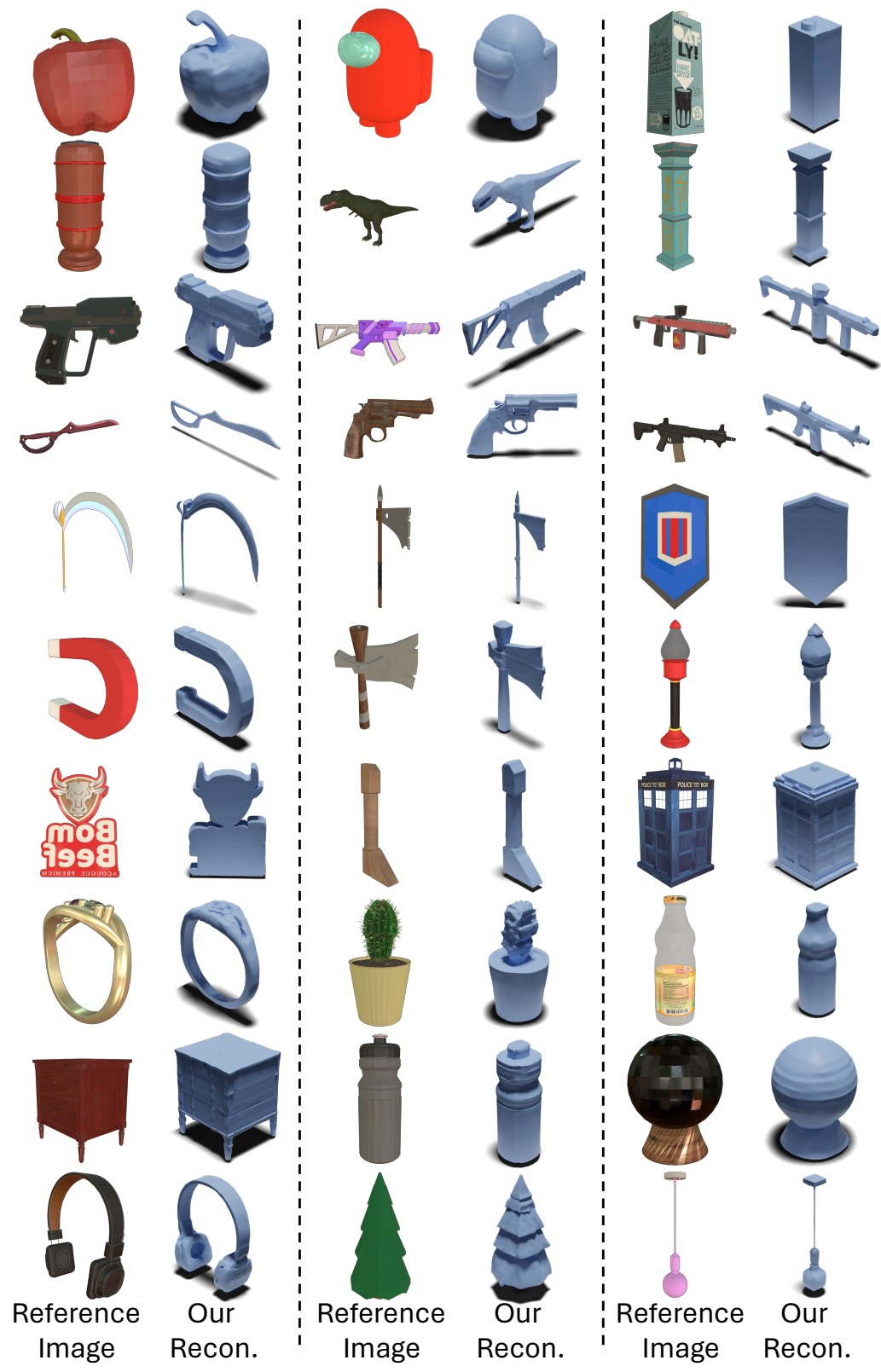

| Reference Image | Our Recon. | Reference Image | Our Recon. | Reference Image | Our Recon. |

Figure 10: Rendered images of shapes reconstructed by our LAM3D from single images on the Objaverse dataset. For each tuple of samples, the left image is the reference image and the right image is the reconstructed geometry.

