# OpenReview forum: "LAM3D: Large Image-Point Clouds Alignment Model for 3D Reconstruction from Single Image"
_NeurIPS.cc/2024/Conference — NeurIPS 2024 poster_

### Official Review · Reviewer_vD6Z · 2024-07-10

**Soundness:** 4
**Presentation:** 4
**Contribution:** 3
**Rating:** 7
**Confidence:** 5

**Summary:**

The paper addresses the problems of multi-view consistency and geometric detail in image-to-3D generation. The proposed method, LAM3D, adopts a two-stage approach for training. In the first stage, the authors train a plane encoder and decoder to compress point clouds into a latent tri-plane representation. In the second stage, to align image and point cloud features within the same feature space, they train the diffusion model. During inference, the image is converted into a latent tri-plane by passing through the trained U-Nets from the second stage, and the features are decoded by the plane decoder trained in the first stage. A final mesh is extracted using the marching cubes algorithm from the reconstructed tri-plane. Experiments demonstrate the effectiveness of the proposed method in 3D generation from a single image, both quantitatively and qualitatively. Additionally, the authors provide an extensive ablation study on different design components.

**Strengths:**

(1) The task of single-view image to 3D generation is extremely challenging and well-motivated.

(2) The proposed method addresses the limitations of existing baselines, which only use images for training and fail to effectively reflect multi-view consistency and geometric detail. To solve the problems, the authors use point clouds as a 3D prior and converting them into tri-planes to facilitate training. Additionally, the method is carefully designed to align the images and tri-planes within the same feature space.

(3) The results show that the proposed method achieves SOTA performance on single image-to-3D generation with a short inference time compared to existing baselines.

(4) A numerous ablation studies on design choice is conducted to achieve optimal performance.

**Weaknesses:**

(1) As mentioned in the limitation, LAM3D cannot generate a textured mesh, while baselines (One-2-3-45, LRM, and CRM) can. In generating 3D content, the accurate geometry of the mesh is important, but the corresponding texture is also a crucial component. It is unfortunate that this aspect cannot be generated. Although it would take more time, generating multi-view images based on the input image using a multi-view diffusion model like Zero-1-to-3 and then unprojecting them to the generated mesh could be a solution.

(2) Recently, many papers related to single image-to-3D generation have been published, and there are many papers like [1, 2] that were published 2 months before the NeurIPS 2024 submission deadline. Among these, I guess Michelangelo [1] has the best quality in geometry generation, so comparing this paper would make the argument more compelling.

[1] Zhao et al., Michelangelo: Conditional 3D Shape Generation based on Shape-Image-Text Aligned Latent Representation, NeurIPS 2023

[2] Vikram Voleti et al., SV3D: Novel Multi-view Synthesis and 3D Generation from a Single Image using Latent Video Diffusion, arXiv 2024

**Questions:**

(1) A minor question: does collecting points using KNN and then using the embedding created by PointNet improve performance compared to directly inputting the points extracted through FPS into the transformer?

**Limitations:**

The authors mention that the limitation of the current pipeline cannot generate texture, as discussed in Section 5.

---

> ### Author Rebuttal · Authors · 2024-08-07
>
> Thank you for your insightful questions and suggestions. We have provided our responses below:
>
> ***Q1. Texture***:
>
> We acknowledge the reviewer's concern about LAM3D's current limitation in generating textured meshes, as mentioned in the paper. While our primary focus was on achieving accurate geometry reconstruction, we fully agree that texture is a crucial component in generating realistic 3D content. Our approach was motivated by the need to resolve geometric distortion issues caused by the lack of explicit 3D priors in recent large reconstruction models. We believe that recovering high-quality geometries is foundational for both textured and non-textured 3D reconstruction. We appreciate the suggestion to use a multi-view diffusion model like Zero-1-to-3 to generate textures, and we recognize this as a promising direction to effectively complement LAM3D’s geometry output. Furthermore, we are committed to exploring such solutions in future work to enhance the capabilities of LAM3D.
>
>
> ***Q2. More Comparison***
>
> We appreciate the reviewer highlighting these recent papers, particularly Michelangelo, which also adopts an alignment-based approach for 3D reconstruction. As discussed in response to Q1 of Reviewer *1gWL* and in our main paper, Michelangelo's alignment strategy has two key limitations: (1) the contrastive loss used in Michelangelo enhances linear separability in the latent space, which is beneficial for discriminative tasks but problematic for 3D reconstruction that requires a continuous latent space to capture the morphable nature of 3D objects; (2) the choice of a vector-shaped latent representation falls short in preserving the spatial information necessary for accurate 3D reconstructions. We address these issues with our diffusion-based alignment strategy, which promotes continuity in the latent space, and a tri-plane structured latent representation that retains spatial information. To validate our approach, we constructed a baseline model based on ULIP [58], a framework similar to Michelangelo that unifies 3D, image, and text modalities. Our experiments demonstrated that while Michelangelo's strategy performs well at the ShapeNet scale, it fails to generalize effectively to the large-scale and category-agnostic Objaverse dataset. These results are detailed in Figures 1, 4, and Tab. 2 of our main paper. Additionally, we recognize the promising direction presented by SV3D, which employs large-scale video diffusion models for 3D reconstruction, and we will discuss this in the revised related works section.
>
> ***Q3. KNN+PointNet v.s. FPS+Transformer***
>
> The method mentioned by the reviewer, using KNN for point collection followed by a PointNet encoder, was initially our preferred approach due to its simplicity and efficiency during the algorithm development stage. However, our experiments revealed that the KNN+PointNet combination struggled to fully capture the input geometry, often resulting in oversmoothed reconstructions and low-quality details. This limitation likely arises because the projection operation aggregates information from a broader range of spatial locations, whereas KNN+PointNet is constrained to local operations. This finding motivated our shift to using a Transformer-based approach, which better integrates long-range interactions between local patches, and our experiments demonstrated the effectiveness of this method in improving reconstruction quality.

---

> > ### Comment · Reviewer_vD6Z · 2024-08-12
> > **Response to rebuttal**
> >
> > Thank you for the response. I have read it as well as other reviews.
> >
> > The authors have promised to address the missing implementation details and pipeline figure in a future revision, so there doesn’t seem to be any further cause for concern. Therefore, I will maintain my original rating of 7.

---

> > > ### Author Response · Authors · 2024-08-13
> > >
> > > Thank you for your feedback! As promised, we will refine the implementation details and pipeline figure, as well as any other areas that need improvement. We appreciate your support and consideration.

---

### Official Review · Reviewer_1gWL · 2024-07-11

**Soundness:** 4
**Presentation:** 4
**Contribution:** 3
**Rating:** 5
**Confidence:** 4

**Summary:**

This paper introduces a new 3D generation framework, LAM3D, which uses point cloud data and image-point-cloud feature alignment method to improve the geometry of the generation results. Authors use triplanes as representation, combining image feature as well as point cloud feature. For point cloud compression, they use point cloud transformer to encode the point cloud to triplane. For better generation, they encode the point cloud triplane to latent triplane, and trained three diffusion model to generate latent triplane. The whole method can generate untextured mesh within 6 seconds, showing better geometry results comparing baselines.

**Strengths:**

1. The paper is well-written with good soundness. Logic is clear and motivation is reasonable.
2. Qualitative and quantitative experiment are carried out in detail and are executed with precision. Ablation study was done with a high degree of accuracy, confirming the significance of parallel diffusion and latent triplane.
3. To handle the difficulty of image triplane and point cloud triplane alignment, LAM3D use a plain encoder to encode the initial triplane to latent triplane, this may give the community another insight in 3D representation and 2D image alignment.

**Weaknesses:**

1. There are already some work on Shape-Image-Aligned generation work such as 3DGen[1] and Michelangelo[2], authors should compare with these work and point out what are the advantages of latent triplane and three parallel diffusion model.
2. Parallel diffusion design may lose some 3D information.

[1] Gupta, Anchit, et al. "3dgen: Triplane latent diffusion for textured mesh generation." arXiv preprint arXiv:2303.05371 (2023).

[2] Zhao, Zibo, et al. "Michelangelo: Conditional 3d shape generation based on shape-image-text aligned latent representation." Advances in Neural Information Processing Systems 36 (2024).

**Questions:**

1. For the parallel diffusion part, why not try using one diffusion to output a tensor of size 2x32x96, then resize it to 3x2x32x32 (similar to the design in CRM)? What are the differences, and what are the advantages of parallel diffusion?
2. You could show more results on real-world data, such as MvImgNet[1] or photos taken by phones.
3. Some works based on DiT[2], like Sora and the concurrent work Direct3D[3], have shown great success. Have you tried using other diffusion models?
4. If LAM3D is open-sourced, it could greatly benefit the community. What are your plans for open-sourcing it?

[1] Yu, Xianggang, et al. "Mvimgnet: A large-scale dataset of multi-view images." Proceedings of the IEEE/CVF conference on computer vision and pattern recognition. 2023.

[2] Peebles, William, and Saining Xie. "Scalable diffusion models with transformers." Proceedings of the IEEE/CVF International Conference on Computer Vision. 2023.

[3] Wu, Shuang, et al. "Direct3D: Scalable Image-to-3D Generation via 3D Latent Diffusion Transformer." arXiv preprint arXiv:2405.14832 (2024).

**Limitations:**

Yes.

---

> ### Author Rebuttal · Authors · 2024-08-07
>
> Thank you for your insightful questions and suggestions. We have provided our responses below:
>
> ***Q1. Comparison***
>
> We thank the reviewer for highlighting these related works. Shape-Image-Alignment is not a novel concept, and has been explored in works like Michelangelo and ULIP [58]. The essence of both Michelangelo and ULIP is training a 3D autoencoder with a CLIP-aligned latent space. However, the experiments of Michelangelo were limited to the ShapeNet scale. To examine the effectiveness of this approach on a larger scale, we built a baseline model using the 3D encoder of ULIP, as described in lines 293 to 304, and found notable limitations when scaling to the Objaverse dataset with 140k category-agnostic objects. Firstly, the contrastive loss used in Michelangelo and ULIP enhances linear separability in the latent space, which is beneficial for discriminative tasks. However, this approach is problematic for 3D reconstruction, as it requires a continuous latent space to capture the morphable nature of 3D objects. While Michelangelo succeeded within the relatively small and dense ShapeNet dataset, this alignment approach failed to generalize at the Objaverse scale, leading to poor 3D reconstruction quality as demonstrated in Fig. 1 and Tab. 2. Secondly, the constraint of encoding 3D shapes into a vector feature representation of size $\mathbb{R}^{512}$ to align with pre-trained CLIP features proved inadequate for capturing the rich diversity of 3D geometries, as seen in our comparisons in Fig. 4. In contrast, our diffusion-based alignment strategy fosters a continuous latent space, while our tri-plane latent representation effectively preserves spatial information that is often lost in vector-based latent representations, thereby enabling more accurate and scalable 3D reconstructions. Regarding the 3DGen method, it treats the image as a condition for the diffusion model and uses a CLIP image encoder to produce a vector-shaped feature for the diffusion process. However, as we discussed earlier, this approach can lead to a loss of spatial information. In contrast, our method utilizes DINO as the image encoder, generating per-patch encodings of size $\mathbb{R}^{1025 \times 768}$, which allows the subsequent diffusion alignment module to transform these into a tri-plane representation, preserving crucial spatial information. Additionally, our diffusion model differs fundamentally from 3DGen’s approach, which we discuss further in the response to Q2.
>
> ***Q2. Parallel Diffusion***
>
> We appreciate the reviewer's concern about the potential loss of 3D information. Previous methods often handle tri-plane structures through either roll-out operations, as seen in 3DGen, or channel concatenation, as discussed in response to Q3. However, both approaches have significant drawbacks. The roll-out method involves convolution operations that span adjacent planes, but the values at the edges of these planes are not spatially continuous, leading to interference and inaccuracies during the learning process. Similarly, channel concatenation aligns features at the same spatial location across planes, but without an explicit relationship between them, applying a single convolution kernel can cause interference in the regression of each plane. To address these issues, our method leverages the planar properties of the tri-plane structure by processing each plane independently. This approach effectively resolves inter-plane interference and results in better reconstruction outcomes. Furthermore, the three planes can be viewed as orthographic projections of the 3D object. We explicitly decouple features across planes by introducing a latent tri-plane loss, which preserves the tri-plane structure in the latent space, as shown in Fig. 6 of the appendix. Although our three independent diffusion models may lack direct inter-plane interaction, this design diminishes inter-plane interference and ensures that 3D information is not lost, as the plane features are decoupled and maintained through the latent tri-plane loss during the compression stage.
>
> ***Q3. Channel Concatenation and Parallel Diffusion***
>
> As explained in our response to Q2, using channel concatenation results in the regression of values based on all three planes at the same spatial location, despite there being no explicit relationship between them. This lack of spatial relationship leads to interference during the learning process. To address this, we constructed a baseline model with a single diffusion network that concatenates the planes channel-wise. Our experimental results, presented in Fig. 5 and Tab. 3, demonstrate that the parallel diffusion approach outperforms this single UNet variant, offering better reconstruction quality and visual effects. The parallel diffusion design mitigates the inter-plane interference, leading to more accurate and effective 3D reconstructions.
>
> ***Q4. More Evaluation***
>
> We appreciate the reviewer's suggestion to include results on real-world data. We evaluated our method on the Google Scanned Objects dataset, which includes a diverse range of real-world household objects, and our approach outperforms previous methods as shown in Tab. 1. In response to the reviewer's comment, we have now included additional comparisons on more diverse sources of images in Fig. 3 of the global response.
>
> ***Q5. DiT***
>
> Due to limited computational resources during the development stage, we focused on validating our approach with a parallel UNet design, which delivered state-of-the-art results. However, DiT is indeed a promising direction, and we plan to explore it in future work.
>
> ***Q6. Open Source***
>
> We appreciate the reviewer's interest in open-sourcing of LAM3D. We are actively seeking institutional approval to release both the training and inference codebase to the public. Additionally, we plan to provide an online demo to facilitate quick and easy access to our model for the community.

---

> > ### Comment · Reviewer_1gWL · 2024-08-12
> >
> > Thank you for your response. But I found that the mesh generated by your method does not resemble the original image(such as in your rebuttal file Figure 3 Line 3, the bottle). I have no more questions and I will keep my rating. Thank you for your work!

---

> > > ### Author Response · Authors · 2024-08-13
> > >
> > > Thank you for your feedback! We acknowledge that our method may present some mismatches between the reconstruction and the reference image in certain details. This occurs because our approach uses a diffusion process to align image features with latent tri-plane features, which introduces some statistical variation and can lead to discrepancies in finer details. However, our method effectively addresses the geometry distortion problem and accurately reconstructs the overall structure from the reference image. We appreciate your insights and thank you again for your valuable comments.

---

### Official Review · Reviewer_XUip · 2024-07-14

**Soundness:** 3
**Presentation:** 3
**Contribution:** 2
**Rating:** 4
**Confidence:** 5

**Summary:**

This paper introduces the Large Image and Point Cloud Alignment Model (LAM3D), a novel framework that enhances 3D mesh reconstruction from single images by utilizing point cloud data. LAM3D integrates a point-cloud-based network for generating precise latent tri-planes, followed by an Image-Point-Cloud Feature Alignment technique that enriches image features with robust 3D information. This approach allows for the production of high-fidelity 3D meshes while mitigating geometric distortions commonly associated with single-image inputs. The method demonstrates its effectiveness across various datasets, achieving state-of-the-art performance in high-fidelity 3D mesh reconstruction.

**Strengths:**

1. The paper presents a method combining point cloud data with image features to reconstruct 3D meshes, which is a promissing solution to the problem of geometric inaccuracies in single-image 3D reconstructions.
2. The experimental results are robust, showing clear improvements over existing methods.

**Weaknesses:**

1. The LAM3D framework's omission of texture mapping is a significant limitation, particularly when compared against methods like One-2-3-45 and CRM that integrate both geometric and texture reconstructions. This absence not only restricts the visual and practical applicability of the generated meshes but also raises concerns about the fairness of comparisons made in the paper. The inclusion of texture details is pivotal for realistic 3D reconstructions, and its absence in LAM3D suggests a crucial area for potential improvement. Furthermore, the comparisons with models that handle both geometry and texture may not provide a balanced view. It's more appropriate to compare LAM3D against models such as "Slice3D: Multi-Slice, Occlusion-Revealing, Single View 3D Reconstruction" (CVPR, 2024) and "D^2IM-Net: Learning Detail Disentangled Implicit Fields from Single Images" (CVPR, 2021), which are more aligned in terms of focusing primarily on geometric details.
2. LAM3D requires point cloud data during training, its applicability is notably restricted to specific categories or types of objects. This limitation indicates a constrained generalization ability, which may impede its deployment in diverse real-world scenarios that demand robust 3D reconstruction capabilities across varied object environments.
3.  The use of independent diffusion processes for each tri-plane is a novel approach, but the paper does not provide a comparative analysis of this method against more integrated approaches.

**Questions:**

1. Can you provide more details on how LAM3D handles various object categories, especially those not directly represented in the training datasets?
2. Could the authors elaborate on how LAM3D might be adapted or extended to include texture reconstruction?

**Limitations:**

While LAM3D effectively enhances mesh reconstruction quality by leveraging 3D point cloud data, its current implementation omits texture mapping. This omission not only limits the visual realism and practical utility of the generated 3D meshes but also makes direct comparisons with methods like One-2-3-45 and CRM, which include textural details, somewhat unfair.

---

> ### Author Rebuttal · Authors · 2024-08-07
>
> Thank you for your insightful questions and suggestions. We have provided our responses below:
>
> ***Q1. Omission of Texture and Texture Extension***
>
> We appreciate the reviewer's observation regarding the omission of texture mapping. Our research primarily focused on addressing the limitations of previous methods that rely solely on volumetric rendering for training large-scale 3D reconstruction models, often neglecting the integration of explicit 3D prior knowledge. By aligning image domain features with the 3D domain, our approach significantly improves geometry reconstruction capabilities.
>
> While texture mapping is left for future work, our method can be easily extended to reconstruct both texture and geometry. This would involve modifying the first stage of training by adjusting the autoencoder to accept a colored point cloud, then introducing an additional MLP to decode the interpolated tri-plane features and regress an RGB vector. In this extended approach, the model would decode both the signed distance and the color at corresponding positions. The inference pipeline remains unchanged, as we can use Marching Cubes to extract both geometry and texture. Another potential approach, as suggested by Reviewer *vD6Z*, involves unprojecting multi-view images generated from models like zero-1-to-3.
>
> ***Q2. Evaluation Fairness***
>
> We appreciate the reviewer’s concern regarding fairness. We believe our comparison is fair because our primary aim is to address the geometry distortion problem caused by the lack of explicit 3D prior knowledge in large reconstruction models. Our method tackles a fundamental aspect of 3D modeling that is crucial for both textured and non-textured models. Additionally, we have included a comparison with the Slice3D model in our global response (Fig. 2 and Fig. 3) to ensure our evaluation covers methods with a similar geometric focus. While D$^2$IM-Net is a pioneering work in disentangling detail in implicit fields, it was trained on ShapeNet, a smaller and less diverse dataset than the large-scale Objaverse dataset we used, making a direct performance comparison challenging. We will discuss both Slice3D and D$^2$IM-Net in the related work section.
>
> ***Q3. Constrained Generalization Ability***
>
> We appreciate the reviewer's observation regarding the requirement for point cloud data during training. Unlike previous rendering-based reconstruction models like LRM and CRM that utilize both 3D assets and multi-view images for training, our approach is constrained to train solely on 3D assets. Despite this, our method demonstrates strong generalization capabilities. While LRM uses 730k 3D assets and 220k videos and CRM employs 376k 3D objects for training, our model is trained on a smaller, category-agnostic dataset of 140k 3D assets. Nonetheless, we achieve state-of-the-art performance on the GSO dataset, which contains over 1,000 3D-scanned real-world items not included in our training data. This highlights the effectiveness of our method in generalizing across varied objects. We argue that although rendering-based methods benefit from larger datasets, their effectiveness is often limited by the lack of explicit 3D knowledge, which can reduce data efficiency. In contrast, our approach, with direct access to 3D representations, enables more efficient learning and improves our ability to generalize across diverse object categories.
>
> ***Q4. Diffusion***
>
> Our motivation for using independent diffusion UNets is outlined in lines 51-53 of the paper, and we have experimentally demonstrated that the design leads to improved alignment quality (Tab. 3 and Fig. 5 of our paper). The UNet-based diffusion process is inherently designed to process 2D planar data, while tri-plane are composed of three 2D orthographic projections of a 3D object. Previous methods, such as 3DGen [11] and NFD [43], either roll out the tri-plane as a continuous plane or concatenate channels based on spatial positions, both of which have significant drawbacks. The roll-out method involves convolution operations that overlap between planes, yet the values at the edges of adjacent planes are not spatially continuous, leading to interference. Conversely, while channel concatenation aligns features at the same spatial location, there is no explicit relationship between them, and applying the same convolution kernel to process them results in interference with the regression of each plane. These issues and their impact on convolution operations are illustrated in Fig. 1 of the global response.
>
> To address these problems, we leverage the tri-plane structure’s property of being three orthographic representations of an object and employed independent diffusion processes for each plane to effectively align image features with planar features. Our ablation study, detailed in lines 305 to 314 of our paper, further supports the effectiveness of this approach. We will also include this additional analysis of the independent diffusion method in the revised paper.
>
> ***Q5. Various Categories***
>
> Unlike methods trained on ShapeNet in a category-specific manner with a limited number of object categories, our approach is category-agnostic and does not rely on predefined categories during training. This strategy enables our model to handle a broader range of objects by focusing on general geometric features rather than specific types. Similar to other large reconstruction models, our training data includes a wide variety of object shapes and structures. To validate our model's generalization capability, we conducted evaluations on the GSO dataset, which is composed of objects not seen during training, as shown in Fig. 3 of our paper. Our model not only successfully handled these unseen objects but also outperformed existing methods, demonstrating robustness across different categories. We also provide supplementary evaluations on data from various sources in Fig. 3 of the global response.

---

### Official Review · Reviewer_tycK · 2024-07-15

**Soundness:** 3
**Presentation:** 3
**Contribution:** 3
**Rating:** 4
**Confidence:** 2

**Summary:**

The paper proposes a two-stage 3D reconstruction method that first uses a transformer-based 3D point cloud feature extractor to initialize hierarchical latent triplanes (XY, XZ, YZ) and reconstructs the 3D mech. Next, it presents an image-point cloud feature alignment approach that leverages initial latent triplanes to align them with the image-based features using independent plane diffusion models to produce better detail and reduced geometric distortion reconstructions.

The approach achieves state-of-the-art high-fidelity 3D mesh reconstructions from a single image in just 6 seconds, and experiments on various datasets demonstrate its effectiveness.

**Strengths:**

•	The concept of utilizing the proposed image-point cloud feature alignment approach using an independent plane diffusion model seems distinctive.
•	The proposed method of 3D reconstructions shows reduced geometric distortion reconstructions compared to the currently mentioned methods.
•	Achieves SOTA results on the mentioned 3D object-based dataset.

**Weaknesses:**

•	Comparison of the model capacity with the current SOTA methods might be added as the proposed methods seems heavy that uses three diffusion, three UNet and two transformer-based architecture.
•	Also, the paper might want to compare the inference time with the existing methods as it specifies that it only requires 6 secs for inference.
•	Details of the module and inference stage missing (see section correctness).
•	The method is evaluated using only a single dataset.
•	Lack of reproducibility

The paper is not well-written. The figure is supposed to give an overview of the pipeline, however, if just reading the figure it is not clear what the input is. The paper refers to input point clouds multiple times, which makes it furthermore confusing.  Or maybe the point clouds are also part of the input as it is referred to as prior point clouds in the paper. If so the performance gain might also be due to the additional multi-modality inputs which is not a fair comparison with the other baseline methods.

Typo:

•	Line 622: We follow LRM [14] ‘and uses’ a 12-layer transformer to directly align image feature to point cloud seems mistaken.

**Questions:**

The paper needs to be improved in the presentation and make the flow chart more clear for example, what is the input?

•	Missing details of the Point Net and Transformer block utilized in the point feature extraction stage, plane encoder in triplane compression, UNet utilized for the plane refiner process, and diffusion models in image-point cloud alignment.
•	The inference process is unclear, specifically lines 249-251, and might need more discussion.
•	The paper misses mentioning the concept and gaps of the current compared methods called One-2-3-45 [24], SyncDreamer [25], TGS [63], CRM [51], and Magic123 [35] in the related work.

**Limitations:**

The authors adequately addressed the limitations.

Overall, the idea of aligning the 3D point features induced triplane representation and image-based features utilizing independent plane diffusion models seems interesting and can produce less geometric distortion reconstruction. However, the proposed method is evaluated only on a single dataset and misses the related work, mentioned module details, and model capacity comparison that must be addressed.

---

> ### Author Rebuttal · Authors · 2024-08-07
>
> Thank you for your insightful questions and suggestions. We have provided our responses below:
>
> ***Q1. Model Capacity***
>
> As shown in Tab. 1 of the global response, we provide parameter size comparisons between our model and SOTA methods. It is worth noting that recent large reconstruction models are designed with high capacity to leverage large-scale training data, enhancing their generalizability in reconstructing novel objects. Our method aligns with this design philosophy.
>
> ***Q2. Inference Time***
>
> As shown in Tab. 1 of the global response, our method achieves better inference time. Specifically, our time breakdown is as follows: approximately 0.02 seconds for encoding an image with DINO, 4.52 seconds to align the image feature to latent tri-planes using 50 diffusion steps, 0.24 seconds to upsample the latent tri-plane to the raw tri-plane space with the plane decoder and refiner, and 1.05 seconds for mesh extraction using Marching Cubes.
>
> ***Q3. Missing Details***
>
> We have described the design of each module in the method section. We mainly focus on the conceptual aspects in our paper, and we acknowledge that the implementation details are somewhat limited due to page limitations. The inference pipeline is outlined in the caption of Fig. 2 and in Section 4.1, lines 249 to 251. We will provide additional explanations to clarify any potential misunderstandings.
>
> ***Q4. Single Dataset Evaluation***
>
> Our evaluation protocol was precisely adopted from CRM (ECCV 2024) [51]. A common practice in large reconstruction models is to train on the Objaverse dataset and evaluate on the Google Scanned Objects (GSO) dataset. This approach was also used in TGS (CVPR 2024) [63] and One-2-3-45 (NeurIPS 2023) [24]. In response to the reviewer's feedback, we have included additional evaluation results in Fig. 3 of the global response. These results encompass reconstruction outcomes for images from ImageNet, MVImageNet, AI generated images, and images collected from the Internet.
>
> ***Q5. Clarity***
>
> We appreciate the reviewer highlighting areas for improvement. We acknowledge the concerns regarding the clarity of the figure and the input description in our manuscript. To clarify, our pipeline consists of two stages during training: point clouds are used as input for the first stage, while images are used for the second stage. During inference, only images are required as input. Thus, the performance gains achieved are attributed to the effectiveness of our proposed modules and strategies, rather than the use of additional input modalities. We will revise the figure and manuscript to clearly distinguish between the training and inference phases, ensuring that the role of each input is clearly communicated.
>
> ***Q6. Presentation***
>
> We appreciate the reviewer’s feedback on the presentation and clarity of the flow chart. We recognize that the current chart may not clearly indicate the inputs. We will revise the flow chart to explicitly show that point clouds are used in the first training stage, images in the second training stage, and only images are needed during inference. We will ensure that the inputs are clearly highlighted at each stage of the pipeline to improve overall clarity.
>
> ***Q7. Lack of Details and Reproducibility***
>
> In our manuscript, we focused on the conceptual aspects and design choices of each module to provide a clear understanding of the overall framework and its motivations. We acknowledge that this approach did not cover the implementation details of each component in depth. To address this, we will include additional implementation details in the appendix. Additionally, we are actively seeking institutional approval to release code and pre-trained models to the public for easy reproduction of our results by the community.
>
> ***Q8. Inference Process***
>
> We apologize for any confusion caused by the description of the inference process in lines 249-251. To clarify, our model requires only a single-view image of an object during inference. The process starts by encoding this image into a feature representation using the DINO image encoder. This image feature is then aligned to a latent tri-plane representation through three independent diffusion UNets, where the latent tri-plane represents a compressed tri-plane shape in the latent space. The latent tri-plane is subsequently upsampled to a high-resolution tri-plane using the plane decoder and refiner from the first training stage. We then employ the SDF MLP to decode the signed distance for any query position on the tri-plane, as described in lines 173-175, which allows us to use the Marching Cubes algorithm to reconstruct a mesh from the tri-plane. We will revise the manuscript to include a more detailed explanation of this inference process to enhance clarity.
>
> ***Q9. Concept and Gaps***
>
> We appreciate the feedback. We will expand the related work to include a comprehensive review of [24,25,63,51,35]. Specifically, we will highlight that these previous methods primarily employ a rendering-based loss for supervision, which overlooks the use of available explicit 3D priors. This absence of explicit 3D priors often results in 3D geometric distortions, examples of which are demonstrated in Fig. 1(b) for LRM [12] and Fig. 1(c) for CRM [51]. In contrast, our method leverages 3D point clouds as explicit guidance during the training stage. We explore the approach of achieving accurate single-image 3D reconstruction by aligning image features with explicit 3D features, marking a novel advancement in the domain.
>
> ***Q10. Typo***
>
> We apologize for the oversight. This typo will be corrected in the revised manuscript.

---

### Author Rebuttal · Authors · 2024-08-07

We thank the reviewers for their thoughtful comments and the time invested in reviewing our work. We take all suggestions seriously and are committed to carefully revising the paper based on your feedback.

In the attached PDF, we have included one table and three figures for your reference:

+ **Table 1**: Comparison of model capacity and inference time.

+ **Figure 1**: Illustration of unintended convolution behavior in different methods for processing tri-plane structured data.

+ **Figure 2**: Qualitative comparisons of our method and Slice3D on the Objaverse and GSO datasets.

+ **Figure 3**: Shapes reconstructed by our LAM3D from single images, compared with state-of-the-art methods.

We respond to reviewers' specific concerns individually below.

---

### Decision · Program_Chairs · 2024-09-25

**Decision:**

Accept (poster)

**Comment:**

This paper was reviewed by four experts in the field.  Based on the reviewers' feedback, the decision is to recommend the paper for acceptance.  The reviewers did raise some valuable concerns that should be addressed in the final camera-ready version of the paper. The authors are encouraged to make the necessary changes to the best of their ability.   In particular, it is strongly recommended to release the code as promised in "5. Open access to data and code" field.